

# Characterization and function of medium and large extracellular vesicles from plasma and urine by surface antigens and Annexin V

Ko Igami[1,2,3], Takeshi Uchiumi[1,4], Saori Ueda[1], Kazuyuki Kamioka[2,5], Daiki Setoyama[1], Kazuhito Gotoh[1], Masaru Akimoto[1], Shinya Matsumoto[1] and Dongchon Kang[1]

[1] Department of Clinical Chemistry and Laboratory Medicine, Kyushu University Graduate School of Medical Sciences, Kyushu University, Fukuoka, Japan
[2] Kyushu Pro Search Limited Liability Partnership, Fukuoka, Japan
[3] Business Management Division, Clinical Laboratory Business Segment, LSI Medience Corporation, Tokyo, Japan
[4] Clinical Chemistry, Division of Biochemical Science and Technology, Department of Health Sciences, Faculty of Medical Sciences, Kyushu University, Fukuoka, Japan
[5] Department of Medical Solutions, LSI Medience Corporation, Tokyo, Japan

Corresponding author
Takeshi Uchiumi,
uchiumi@cclm.med.kyushu-u.ac.jp

## ABSTRACT

**Background.** Extracellular vesicles (EVs) are released by most cell types and are involved in multiple basic biological processes. Medium/large EVs (m/lEVs), which are of a different size from exosomes, play an important role in the coagulation in blood, and are secreted from cancer cells, etc., suggesting functions related to malignant transformation. The m/lEVs levels in blood or urine may help unravel pathophysiological findings in many diseases. However, it remains unclear how many naturally-occurring m/lEV subtypes exist as well as how their characteristics and functions differ from one another.

**Methods.** We used the blood and urinal sample from each 10 healthy donors for analysis. Using a flow cytometer, we focus on characterization of EVs with large sizes (>200 nm) that are different from exosomes. We also searched for a membrane protein for characterization with a flow cytometer using shotgun proteomics. We then identified m/lEVs pelleted from plasma and urine samples by differential centrifugation and characterized by flow cytometry.

**Results.** Using proteomic profiling, we identified several proteins involved in m/lEV biogenesis including adhesion molecules, peptidases and exocytosis regulatory proteins. In healthy human plasma, we could distinguish m/lEVs derived from platelets, erythrocytes, monocytes/macrophages, T and B cells, and vascular endothelial cells with more than two positive surface antigens. The ratio of phosphatidylserine appearing on the membrane surface differed depending on the cell-derived m/lEVs. In urine, 50% of m/lEVs were Annexin V negative but contained various membrane peptidases derived from renal tubular villi. Urinary m/lEVs, but not plasma m/lEVs, showed peptidase activity. The knowledge of the new characteristics is considered to be useful as a diagnostic material and the newly developed method suggests the possibility of clinical application.

## INTRODUCTION

Extracellular vesicles (EVs) play essential roles in cell–cell communication and are diagnostically significant molecules. EVs are secreted from most cell types under normal and pathophysiological conditions (*Iraci et al., 2016*; *Ohno, Ishikawa & Kuroda, 2013*). These membrane vesicles can be detected in many human body fluids and are thought to have signaling functions in interactions between cells. Analysis of EVs may have applications in therapy, prognosis, and biomarker development in various fields. The hope is that, using EV analysis, clinicians will be able to detect the presence of disease as well as to classify its progression using noninvasive methods such as liquid biopsy (*Boukouris & Mathivanan, 2015*; *Piccin, 2014*; *Piccin et al., 2015*; *Piccin et al., 2017a*; *Piccin et al., 2017b*; *Piccin, Van Schilfgaarde & Smith, 2015*).

Medium/large extracellular vesicles (m/lEVs) can be classified based on their cellular origins, biological functions and biogenesis (*El Andaloussi et al., 2013*). In a broad sense, they can be classified into m/lEVs with diameters of 100–1,000 nm diameter (membrane blebs) and smaller EVs (e.g., exosomes) with diameters of 30–150 nm (*Raposo & Stoorvogel, 2013*; *Robbins & Morelli, 2014*). The m/lEVs are generated by direct outward budding from the plasma membrane (*D'Souza-Schorey & Clancy, 2012*), while smaller EVs (e.g., exosomes) are produced via the endosomal pathway with formation of intraluminal vesicles by inward budding of multivesicular bodies (MVBs) (*Raposo & Stoorvogel, 2013*). In this study, we analyzed the physical characteristics of EVs from 200 nm to 800 nm in diameter, which we refer to as m/lEVs as per the MISEV2018 guidelines (*Thery et al., 2018*).

Recently, the clinical relevance of EVs has attracted significant attention. In particular, m/lEVs play an important role in tumor invasion (*Clancy et al., 2015*). m/lEVs in blood act as a coagulant factor and have been associated with sickle cell disease, sepsis, thrombotic thrombocytopenic purpura, and other diseases (*Piccin, Murphy & Smith, 2007*; *Piccin et al., 2017a*; *Piccin et al., 2017b*). A possible role for urinary m/lEVs in diabetic nephropathy was also reported (*Sun et al., 2012*). In recent years, the clinical applications of exosomes have been developed (*Yoshioka et al., 2014*). However, because characterization of exosomes is analytically challenging, determining the cells and tissues from which exosomes are derived can be difficult. m/lEVs are generated differently from exosomes (*Mathivanan, Ji & Simpson, 2010*) but are similar in size and contain many of the same surface antigens. It is widely hypothesized that complete separation of exosomes and m/lEVs is likely to be a major challenge, and more effective techniques to purify and characterize m/lEVs would be extremely valuable.

In this study, we focused on m/lEVs in plasma and urine, which are representative body fluids in clinical laboratories. We purified for m/lEVs based on differential centrifugation and characterized m/lEVs by flow cytometry and mass spectrometry analysis and described

the basic properties (characterizing surface antigen and orientation of phosphatidylserine and activity of the enzymes) of m/lEV subpopulations in blood and urine.

## MATERIALS AND METHODS

### Antibodies and other reagents

The following monoclonal antibodies against human surface antigens were used in this study: anti-CD5(clone: L17F12), anti-CD15 (clone: W6D3), anti-CD41(clone: HIP8), anti-CD45(clone: HI30), anti-CD59 (clone: p282), anti-CD61 (clone: VI-PL2), anti-CD105 (clone: 43A3), anti-CD146(clone: P1H12), anti-CD235a (clone: HI264), anti-CD10 (clone: HI10a), anti-CD13 (clone: WM15), anti-CD26 (clone: BA5b), anti-CD227 (MUC1) (clone: 16A). All antibodies were purchased from BioLegend® (San Diego, CA). FITC-conjugated Annexin V was purchased from BD Biosciences (New Jersey, USA). We used the SPHERO™ Nano Fluorescent Particle Size Standard Kit, Yellow (diameters 0.22, 0.45, 0.88 and 1.35 μm) from Spherotech Inc. for size validation. Normal mouse IgG was purchased from Wako Chemicals (Tokyo, Japan). APC-conjugated normal mouse IgG was produced using the Mix-n-Stain™ APC Antibody Labeling kit from Biotium Inc. Dithiothreitol (DTT) was purchased from Wako Chemicals (Tokyo, Japan). We conducted phase transfer surfactant experiments using "MPEX PTS Reagents for MS" purchased from GL Sciences Inc. and "Trypsin, TPCK Treated" purchased from Thermo Fisher Scientific™. Iodoacetamide was purchased from Wako Chemicals (Tokyo, Japan).

### Samples

All studies were approved by the Institutional Review Board of the Kyushu University Hospital, Kyushu University (29-340). Blood samples were collected from 20 male and female participants (23–48 years of age) who were apparently healthy. We received the informed consent from all participants of this study. Samples were collected using a 22-gauge butterfly needle and a slow-fill syringe. After discarding the initial 2–3 mL, blood was dispensed into collection tubes containing ethylenediamine tetra acetic acid (EDTA) (1.6 mg/mL blood). Urine was collected from 20 male healthy subjects (23–46 years of age). The first morning void urine was used for the experiments. The urine samples were collected in a sterile container. In particular, we confirmed that sample used for analysis by flow cytometer were derived from healthy donors by measuring blood count and creatinine in blood and total urine protein (Table S1).

### Isolation of plasma m/lEVs

Essentially platelet-free plasma (PFP) was prepared from EDTA-treated blood by double centrifugation at $2,330 \times g$ for 10 min. To assess residual platelets remaining in this sample, we measured platelet number using the ADVIA® 2120i Hematology System (SIEMENS Healthineers, Erlangen Germany). The number of platelets in this sample was below the limit of detection ($1 \times 10^3$ cells/μL). We used a centrifugation method to obtain m/lEVs. In an effort to ensure our approach could be applied to clinical testing, we chose a simple and easy method for pretreatment. In an ISEV position paper (*Mateescu et al., 2017*), Thery's group referred to vesicles sedimenting at $100,000 \times g$ as "small EVs"

rather than exosomes, those pelleting at intermediate speed (lower than $20,000 \times g$) as "medium EVs" (including microvesicles and ectosomes) and those pelleting at low speed (e.g., $2000 \times g$) as "large EVs". Because these definitions are less biologically meaningful but more experimentally tractable than previously-applied exosome/microvesicle definitions, we attempted biological characterization through subsequent shotgun and flow cytometry analysis.

In flow cytometric analysis, the volume of PFP used in each assay was 0.6 mL from each donor. In electron microscopy, the volume of PFP used was three mL. Samples were independent and were treated individually prior to each measurement. PFP was centrifuged at $18,900 \times g$ for 30 min in a fixed-angle rotor. The m/lEV pellet obtained after centrifugation was reconstituted by vortex mixing (1–2 min) with an equivalent volume of Dulbecco's phosphate-buffered saline (DPBS), pH 7.4. The solution was centrifuged at $18,900 \times g$ for 30 min again and the supernatant was discarded.

### Isolation of urinary m/lEVs

For isolation of urinary m/lEVs, we modified a urinary exosome extraction protocol (*Fernandez-Llama et al., 2010*). The centrifugation conditions were identical for plasma and urine so that the size and the density of m/lEVs were similar, enabling comparison of plasma and urinary m/lEVs.

In flow cytometric analysis, the volume of urine used for each assay was 1.2 mL from each donor. In electron microscopy, the volume of urine used was 15 mL. Samples were independent and were treated individually prior to each measurement. Collected urine was centrifuged at $2,330 \times g$ for 10 min twice. The supernatant was centrifuged at $18,900 \times g$ for 30 min in a fixed-angle rotor. The m/lEV pellet obtained from centrifugation was reconstituted by vortex mixing (1–2 min) with 0.2 mL of DPBS followed by incubation with DTT (final concentration 10 mg/mL) at 37 °C for 10–15 min. The samples were centrifuged again at $18,900 \times g$ for 30 min and the supernatant was discarded. Addition of DTT, a reducing agent, reduced the formation of Tamm-Horsfall protein (THP) polymers. THP monomers were removed from m/lEVs after centrifugation. DTT-containing DPBS solutions were filtered through 0.1-μm filters (Millipore).

### Flow cytometric analysis of m/lEVs

After resuspending m/lEV pellets in 60 μL of DPBS, we added saturating concentrations of several labelled antibodies, Annexin V and normal mouse IgG and incubated the tubes in the dark, without stirring, for 15–30 min at room temperature. In one case, we added labelled antibodies directly to 60 μL of PFP for staining. We resuspended stained fractions in Annexin V binding buffer (BD Biosciences: 10 mM HEPES, 0.14 mM NaCl, 2.5 mM $CaCl_2$, pH 7.4) for analysis by flow cytometry. DPBS and Annexin V binding buffer were filtered through 0.1-μm filters (Millipore). Flow cytometry was performed using a FACSVerse™ flow cytometer (BD Biosciences). The flow cytometer was equipped with 405 nm, 488 nm and 638 nm lasers to detect up to 13 fluorescent parameters. The flow rate was 12 μL /min. Forward scatter voltage was set to 381, side scatter voltage was set to 340, and each threshold was set to 200. Details of excitation (Ex.) and emission (Em.) wavelengths

as well as voltages described in supplements Fig. Flow cytometry was performed using FACSuite[TM] software (BD Biosciences) and data were analyzed using FlowJo software. The authors have applied for the following patents for the characterization method of m/lEVs isolated from plasma and urine with a flow cytometer: JP2018-109402(plasma) and JP2018-109403(urine).

## Nanoparticle tracking analysis (NTA)

NTA measurements were performed using a NanoSight LM10 (NanoSight, Amesbury, United Kingdom). After resuspending mEV pellets in 50 µL of DPBS, samples were diluted eight-fold (plasma) and 100-fold (urinary) with PBS prior to measurement. Particles in the laser beam undergo Brownian motion and videos of these particle movements are recorded. NTA 2.3 software then analyses the video and determines the particle concentration and the size distribution of the particles. Twenty-five frames per second were recorded for each sample at appropriate dilutions with a "frames processed" setting of 1,500. The detection threshold was set at "7 Multi" and at least 1,000 tracks were analyzed for each video.

## Electron microscopy

For immobilization, we added 100 µL of PBS and another 100 µL of immobilization solution (4% paraformaldehyde, 4% glutaraldehyde, 0.1 M phosphate buffer, pH 7.4) to m/lEV pellets. After stirring, we incubated at 4 °C for 1 h. For negative staining, the samples were adsorbed to formvar film-coated copper grids (400 mesh) and stained with 2% phosphotungstic acid, pH 7.0, for 30 s. For observation and imaging, the grids were observed using a transmission electron microscope (JEM-1400Plus; JEOL Ltd., Tokyo, Japan) at an acceleration voltage of 100 kV. Digital images ($3,296 \times 2,472$ pixels) were taken with a CCD camera (EM-14830RUBY2; JEOL Ltd., Tokyo, Japan).

## Protein digestion

We used approximately 50 mL of pooled healthy plasma and 100 mL of pooled healthy male urine from five healthy subjects for digestion of m/lEVs.

In plasma the following process is the same as "Isolation of plasma m/lEVs" section. We repeated $18,900 \times g$ centrifugation washing steps three times to reduce levels of contaminating free plasma proteins and small EVs for shotgun analysis. After the last centrifugation, we removed supernatants and froze the samples.

In urine the following process is the same as "Isolation of urinary m/lEVs" section. We repeated washing steps twice (after DTT treatment) to reduce levels of contaminating free urinary proteins and small EVs for shotgun analysis. We removed supernatants and froze the samples.

To discover characterizing surface antigen by flowcytometry, the sample was digested using a phase transfer surfactant-aided procedure so that many hydrophobic membrane proteins could be detected (*Chen et al., 2017*). The precipitated frozen fractions of plasma and urine were thawed at 37 °C, and then m/lEVs were solubilized in 250 µL of lysis buffer containing 12 mM sodium deoxycholate and 12 mM sodium lauroyl sarcosinate in 100 mM Tris-HCl, pH 8.5. After incubating for 5 min at 95 °C, the solution was sonicated using

an ultrasonic homogenizer. Protein concentrations of the solutions were measured using a bicinchoninic acid assay (Pierce™ BCA Protein Assay Kit; Thermo Fisher Scientific).

Twenty microliters of the dissolved pellet (30 µg protein) were used for protein digestion. Proteins were reduced and alkylated with 1 mM DTT and 5.5 mM iodoacetamide at 25 °C for 60 min. Trypsin was added to a final enzyme:protein ratio of 1:100 (wt/wt) for overnight digestion. Digested peptides were acidified with 0.5% trifluoroacetic acid (final concentration) and 100 µL of ethyl acetate was added for each 100 µL of digested m/lEVs. The mixture was shaken for 2 min and then centrifuged at $15,600 \times g$ for 2 min to obtain aqueous and organic phases. The aqueous phase was collected and desalted using a GL-Tip SDB column (GL Sciences Inc).

## LC-MS/MS analysis

Digested peptides were dissolved in 40 µL of 0.1% formic acid containing 2% (v/v) acetonitrile and 2 µL were injected into an Easy-nLC 1000 system (Thermo Fisher Scientific). Peptides were separated on an Acclaim PepMap™ RSLC column (15 cm ×50 µm inner diameter) containing C18 resin (2 µm, 100 Å; Thermo Fisher Scientific™), and an Acclaim PepMap™ 100 trap column (two cm ×75 µm inner diameter) containing C18 resin (3 µm, 100 Å; Thermo Fisher Scientific™). The mobile phase consisted of 0.1% formic acid in ultrapure water (buffer A). The elution buffer was 0.1% formic acid in acetonitrile (buffer B); a linear 200 min gradient from 0%–40% buffer B was used at a flow rate of 200 nL/min. The Easy-nLC 1000 was coupled via a nanospray Flex ion source (Thermo Fisher Scientific™) to a Q Exactive™ Orbitrap (Thermo Fisher Scientific™). The mass spectrometer was operated in data-dependent mode, in which a full-scan MS (from 350 to 1,400 m/z with a resolution of 70,000, automatic gain control (AGC) 3E+06, maximum injection time 50 ms) was followed by MS/MS on the 20 most intense ions (AGC 1E+05, maximum injection time 100 ms, 4.0 m/z isolation window, fixed first mass 100 m/z, normalized collision energy 32 eV).

## Proteome data analysis

Raw MS files were analyzed using Proteome Discoverer software version 1.4 (Thermo Fisher Scientific™) and peptide lists were searched against the Uniprot Proteomes-Homo sapiens FASTA (Last modified November 17, 2018) using the Sequest HT algorithm. Initial precursor mass tolerance was set at 10 ppm and fragment mass tolerance was set at 0.6 Da. Search criteria included static carbamidomethylation of cysteine (+57.0214 Da), dynamic oxidation of methionine (+15.995 Da) and dynamic acetylation (+43.006 Da) of lysine and arginine residues.

## Gene ontology analysis and gene enrichment analysis

We conducted GO analysis using DAVID (https://david.ncifcrf.gov) to categorize the proteins identified by shotgun analysis and used Metascape (http://metascape.org/gp/index.html#/main/step1) for gene enrichment analysis. We uploaded the UNIPROT_ACCESSION No. for each protein.

## Extracellular vesicle preparation from isolated erythrocytes

Whole blood was collected by the same method as above and centrifuged at $2,330\times g$ for 10 min. After removal of the buffy coat and supernatant plasma, the remaining erythrocytes were washed three times by centrifugation at $2,330\times g$ for 10 min and the erythrocyte pellet was resuspended in DPBS. EVs were generated from the washed erythrocytes by stimulation in the presence of 2.5 mM $CaCl_2$ (10 mM HEPES, 0.14 mM NaCl, 2.5 mM $CaCl_2$, pH 7.4) for 1 h at room temperature under rotating conditions. Erythrocytes were removed by centrifugation at $2,330\times g$ for 10 min and the EV rich supernatant was subsequently centrifuged ($18,900\times g$ for 30 min) to pellet the EVs. EVs were resuspended in DPBS.

## Dipeptidyl peptidase IV (DPP4:CD26) activity assay

DPP4 activity was measured in the plasma and urine of six individuals (different from plasma donors). The method was previously published in part (*Kawaguchi et al., 2010*). DPP4 activity was measured via the fluorescence intensity of 7-amino-4-methylcoumarin (AMC) after its dissociation from the synthetic substrate (Gly-Pro-AMC •HBr) catalyzed by DPP4. Experiments were performed in 96-well black plates. Titrated AMC was added to each well to prepare a standard curve. Fluorescence intensity was measured after incubating substrate with urine samples for 10 min. The enzyme reaction was terminated by addition of acetic acid. The fluorescence intensity (Ex. = 380 nm and Em. = 460 nm) was measured using Varioskan Flash (Thermo Fisher Scientific™). DPP4 activity assays were performed by Kyushu Pro Search LLP (Fukuoka, Japan).

# RESULTS

## Isolation and characterization of m/lEVs from plasma and urine

The workflow for the isolation and enrichment of m/lEVs for flow cytometric analyses is illustrated in Figs. 1A and 1B. m/lEVs from human plasma samples were isolated by high-speed centrifugation, an approach used in previous studies (*Jayachandran et al., 2012*). For isolation of m/lEVs from urine, DTT, a reducing agent, was used to remove THP polymers because these non-specifically interact with IgGs.

Transmission electron microscopy revealed that almost all m/lEVs were small, closed vesicles with a size of approximately 200 nm that were surrounded by lipid bilayer (Figs. 1C–1H). In plasma, we observed EVs whose membranes were not stained either inside or on the surface (Figs. 1C, 1D); we also observed EVs whose forms were slightly distorted (Fig. 1E). In urine, a group of EVs with uneven morphology and EVs with interior structures were observed (Fig. 1H). Apoptotic bodies, cellular debris, and protein aggregates were not detected.

No EVs with diameters greater than 800 nm were observed by NTA (Fig. S1) and flow cytometry can detect only EVs with diameters larger than 200 nm. Together, these data suggested that we characterized m/lEVs between 200 nm and 800 nm in diameter from plasma and urine by flow cytometry analysis. We observed the m/lEVs less than 100 nm by NTA because of some contamination or degradation after purification (Fig. S1).

Side-scatter events from size calibration beads of (diameters: 0.22, 0.45, 0.88 and 1.35 μm) were resolved from instrument noise using a FACS Verse flow cytometer

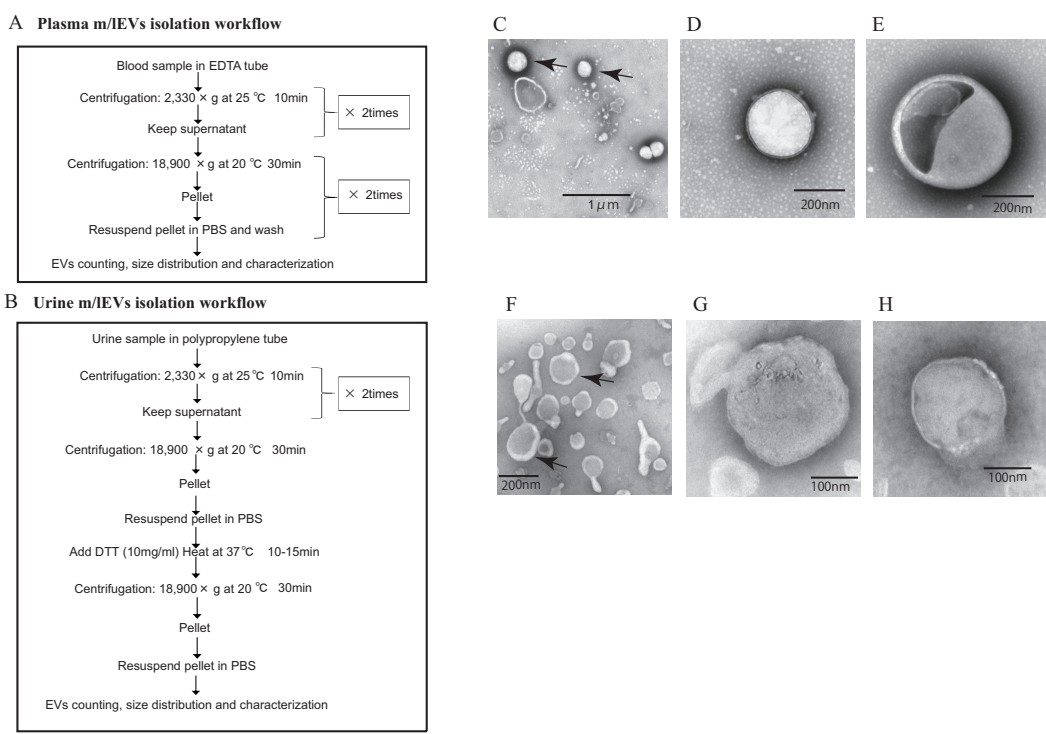

**Figure 1  Isolation of m/lEVs from plasma and urine using differential centrifugation.** (A and B) Workflow of plasma (A) and urine (B) m/lEV isolation and sample preparation for flow cytometry analysis. (C–H) Isolated m/lEVs from plasma (C–E) and urine (F–H) were visualized by transmission electron microscopy. Arrow indicates representative m/lEVs (C and F). Microscopy was used to identify EV-like particles based on the size (100–400 nm) and shape (round) of the vesicles. The scale bar is shown.

(Fig. S2A). Inspection of the side-scatter plot indicated that 0.22 µm was the lower limit for bead detection. More than 90% of m/lEVs isolated from plasma and urine showed side-scatter intensities lower than those of 0.88-µm calibration beads (Figs. 2A–2D). m/lEVs were heterogeneous in size, with diameters ranging from 200–800 nm in plasma and urine (Figs. 2A–2D). Fluorescently-labeled mouse IgG was used to exclude nonspecific IgG-binding fractions (Figs. S2B and S2C). In this experiment, we characterized m/lEVs with diameters ranging from 200–800 nm. NTA analysis shows less than 100 nm size particles in the plasma fraction extracted by centrifugation, but we focused on particles over 200 nm using a flow cytometer. Using these methods, we observed an average of $8 \times 10^5$ and $1 \times 10^5$ m/lEVs in each mL of plasma and urine by flow cytometry observation.

## Shotgun proteomic analysis of plasma and urine EVs

To analyze the protein components and discover characterizing surface antigen of m/lEVs present in plasma and urine of five healthy individuals, we performed LC-MS/MS proteomic analysis. In this analysis, in order to prevent small EVs contamination, the washing process by centrifugation was increased compared to other analyses (Materials and Methods). A total of 593 and 1,793 proteins were identified in m/lEVs from plasma and urine, respectively (Fig. 3A and Tables S2 and S3). Scoring counts using the SequestHT algorithm for the top

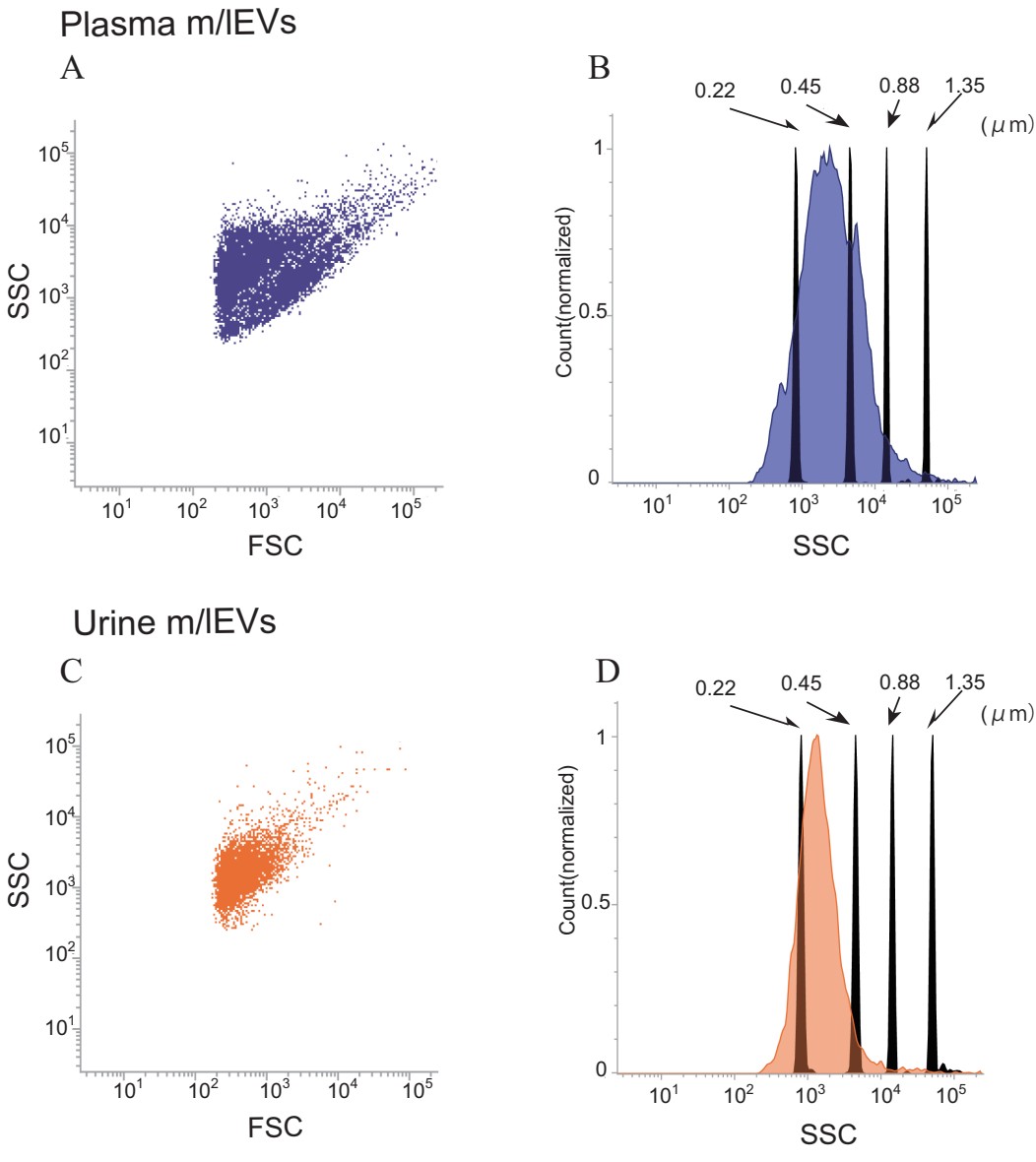

**Figure 2 Flow cytometric analysis of plasma and urine m/lEVs.** (A and B) Analysis of plasma m/lEVs by flow cytometry. Forward and side scatter (SSC) were measured for plasma m/lEVs (A). The SSC distribution of plasma m/lEVs is shown as a histogram (indigo blue) compared with standard polystyrene beads (black histogram) (B). (C and D) Analysis of urine m/lEVs by flow cytometry. Forward and side scatter (SSC) were measured for urine m/lEVs (C). The SSC distribution of urinary m/lEVs is shown as a histogram (orange) compared with standard polystyrene beads (black histogram) (D).

20 most abundant proteins are shown in Tables 1 and 2. We detected cytoskeleton-related protein such as actin, ficolin-3 and filamin and cell–surface antigen such as CD5, band3 and CD41 in plasma. We also identified actin filament-related proteins such as ezrin, radixin, ankylin and moesin which play key roles in cell surface adhesion, migration and organization in both plasma and urine. In urine, several types of peptidases (membrane

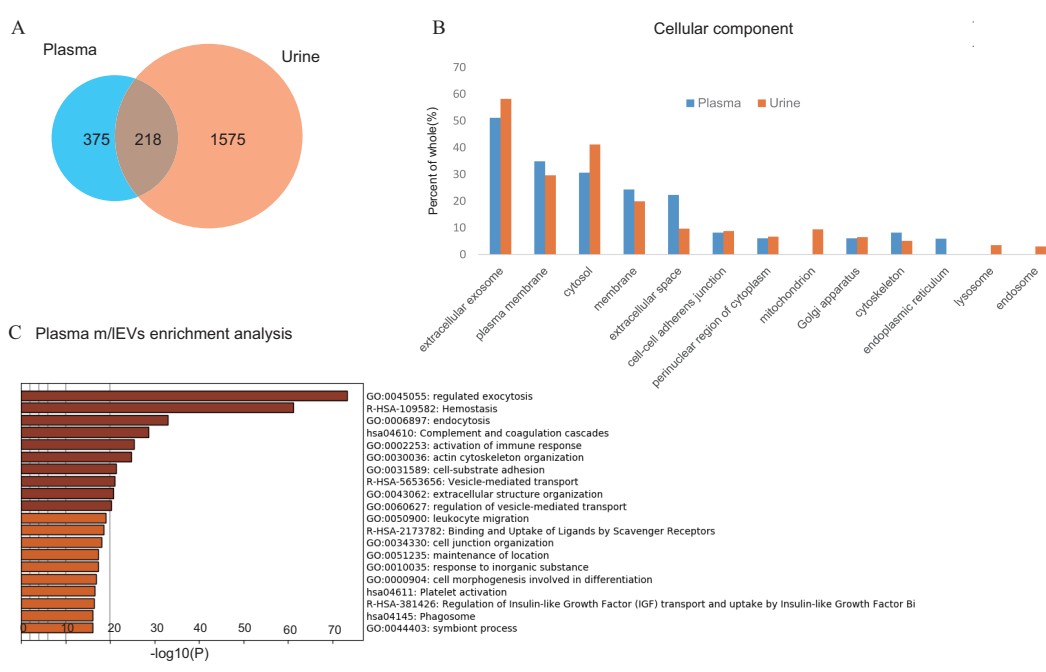

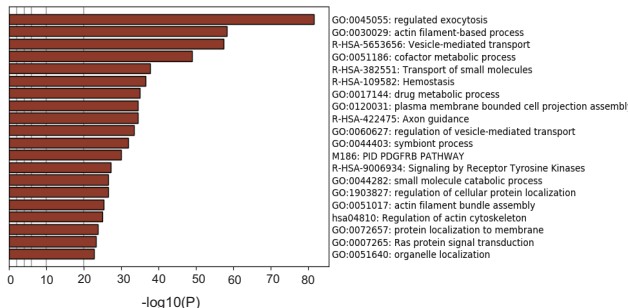

**Figure 3  Shotgun proteomic analysis of plasma and urine m/lEVs.** (A) Protein extracts of m/lEVs isolated from plasma and urine were analyzed by LC-MS/MS. A total of 593 and 1793 proteins from plasma and urine, respectively, were detected. Detailed lists of proteins are shown in Tables S1 and S2. (B) GO (gene ontology) cellular components are shown for m/lEVs isolated from plasma and urine using the DAVID program. Among the detected proteins, the gene list used for DAVID analysis included 588 proteins (plasma) and 1786 proteins (urine). The vertical axis shows the percentage of proteins from the full gene list categorized into each GO term. For example, for extracellular exosomes (plasma), the categorized count was 301 of 588 proteins. (C and D) Top 20 clusters from the Metascape pathway (http://metascape.org/) enrichment analysis for m/lEVs in plasma (C) and urine (D). Lengths of bars represent log10 (*P* values) based on the best-scoring term within each cluster. Among all detected proteins, 535 (plasma) and 1,767 (urine) genes were recognized as unique for enrichment analysis. For each gene list, pathway and process enrichment analysis was carried out using the following ontology sources: KEGG Pathway, GO Biological Processes, Reactome Gene Sets, Canonical Pathways and CORUM.

alanine aminopeptidase or CD13; neprilysin or CD10; DPP4 or CD26) and MUC1 (mucin 1 or CD227) were detected in high abundance, and these proteins were used to characterize m/lEVs by flow cytometric analysis (Table 2 and Table S3). We demonstrated that the isolated m/lEVs showed high expression of tubulin and actinin, while the tetraspanins CD9

**Table 1 Twenty most abundant proteins identified in plasma m/lEVs.** Proteins in bold text indicate antigens identified using flow cytometry. This table excluded immunoglobulin-related proteins and albumin.

| Protein name | UniProt AC | Score | Sequence coverage |
|---|---|---|---|
| Ficolin-3 | O75636 | 8,772 | 68 |
| Hemoglobin subunit alpha | P69905 | 1,735 | 70 |
| Actin, cytoplasmic 1 | P60709 | 927 | 69 |
| Hemoglobin subunit beta | P68871 | 602 | 79 |
| Actin, gamma-enteric smooth muscle | P63267 | 500 | 32 |
| Talin-1 | Q9Y490 | 368 | 50 |
| Filamin-A | P21333 | 353 | 40 |
| Spectrin alpha chain, erythrocytic 1 | P02549 | 321 | 42 |
| Myosin-9 | P35579 | 317 | 38 |
| Mannan-binding lectin serine protease 1 | P48740 | 308 | 35 |
| Band 3 anion transport protein | P02730 | 277 | 38 |
| Beta-actin-like protein 2 | Q562R1 | 252 | 24 |
| Hemoglobin subunit delta | P02042 | 215 | 64 |
| Spectrin beta chain, erythrocytic | P11277 | 183 | 28 |
| Complement C1q subcomponent subunit C | P02747 | 170 | 24 |
| Ankyrin-1 | P16157 | 164 | 28 |
| **CD5 antigen-like** | O43866 | 164 | 52 |
| **Integrin alpha-IIb (CD41)** | P08514 | 129 | 25 |
| Complement C1q subcomponent subunit B | P02746 | 122 | 41 |
| Deaminated glutathione amidase | Q86X76 | 118 | 6 |

and CD81 that are often used as exosome markers were only weakly identified. Especially in plasma, small EV (exosome) markers TSG101, VPS4 and Alix were not observed in this m/lEVs fraction (Table S4). These data suggest that m/lEVs differ from small EVs including exosomes.

As shown in Fig. 3A and Fig. S3, about 10% of urinary EVs proteins were also identified in plasma EVs. Urinary EVs in the presence of blood contaminants were also observed in previous studies (*Smalley et al., 2008*). These result suggest that m/lEVs in plasma were excreted in the urine via renal filtration and not reabsorbed. Gene ontology analysis of the identified proteins indicated overall similar cellular components in plasma and urine m/lEVs (Fig. 3B). The results of gene set enrichment analysis by metascape are shown for plasma and urine m/lEVs (Figs. 3C, 3D and Tables S5 and S6). The most commonly-observed functions in both plasma and urine were "regulated exocytosis", "hemostasis" and "vesicle-mediated transport". In plasma, several functions of blood cells were observed, including "complement and coagulation cascades" and "immune response". Moreover, analysis of urinary EVs showed several characteristic functions including "transport of small molecules", 'metabolic process" and "cell projection assembly". This may reflect the nature of the kidney, the urinary system and tubular villi. These data demonstrate the power of data-driven biological analyses.

**Table 2 Twenty most abundant proteins identified in urinary m/lEVs.** Proteins in bold text indicate antigens identified using flow cytometry.

| Protein name | UniProt AC | Score | Sequence coverage |
|---|---|---|---|
| Actin, cytoplasmic 1 | P60709 | 1,618 | 67 |
| **Neprilysin** | P08473 | 1,194 | 50 |
| Uromodulin | P07911 | 821 | 44 |
| Solute carrier family 12 member 1 | Q13621 | 720 | 32 |
| Alpha-enolase | P06733 | 708 | 79 |
| Moesin | P26038 | 548 | 73 |
| Ezrin | P15311 | 544 | 56 |
| **Aminopeptidase N** | P15144 | 486 | 43 |
| Actin, gamma-enteric smooth muscle | P63267 | 476 | 28 |
| Pyruvate kinase PKM | P14618 | 447 | 64 |
| Voltage-dependent anion-selective channel protein 1 | P21796 | 446 | 74 |
| Radixin | P35241 | 364 | 56 |
| Tyrosine-protein phosphatase non-receptor type 13 | Q12923 | 363 | 38 |
| Triosephosphate isomerase | P60174 | 325 | 80 |
| Multidrug resistance protein 1 | P08183 | 324 | 38 |
| Guanine nucleotide-binding protein G(I)/G(S)/G(T) subunit beta-2 | P62879 | 316 | 50 |
| Guanine nucleotide-binding protein G(I)/G(S)/G(T) subunit beta-1 | P62873 | 306 | 53 |
| V-type proton ATPase catalytic subunit A | P38606 | 298 | 57 |
| Superoxide dismutase [Cu-Zn] | P00441 | 285 | 91 |
| Epidermal growth factor receptor kinase substrate 8-like protein 2 | Q9H6S3 | 277 | 43 |

## Characterization of plasma EVs by flow cytometry

Next, we characterized m/lEVs in plasma by flow cytometry using antibodies against several surface antigens and Annexin V. To eliminate nonspecific adsorption, we excluded the mouse IgG-positive fraction. (Figs. S2B). Eliminating non-specific reactions to antibodies is important in using human body fluids as diagnostic materials for immunological measurements. By adding mouse IgG-APC to the system, we observed accurate flow cytometry image in which specific surface antigens were recognized by following two points: (1) blocking of non-specific reaction sites, (2) gate-out of positive non-specific reaction. We characterized positive m/lEVs using surface antigens detected by shotgun proteomic analysis and Annexin V (Figs. 4A–4L).

To characterize m/lEVs derived from erythrocytes, T and B cells, macrophages/monocytes, granulocytes, platelets and endothelial cells, we selected nine antigens described in Fig. 4A. Two or more antigens CD235a double-positive and CD45-negative m/lEVs were classified as erythrocyte-derived m/lEVs (Fig. S4B). We confirmed that m/lEVs isolated from erythrocytes in vitro and erythrocytes derived m/lEVs from plasma are characterized by CD59 and CD235a double-positive and CD45-negative (Figs. S5). Determined positive area by addition of EDTA (Figs. S2D), we also show Annexin V

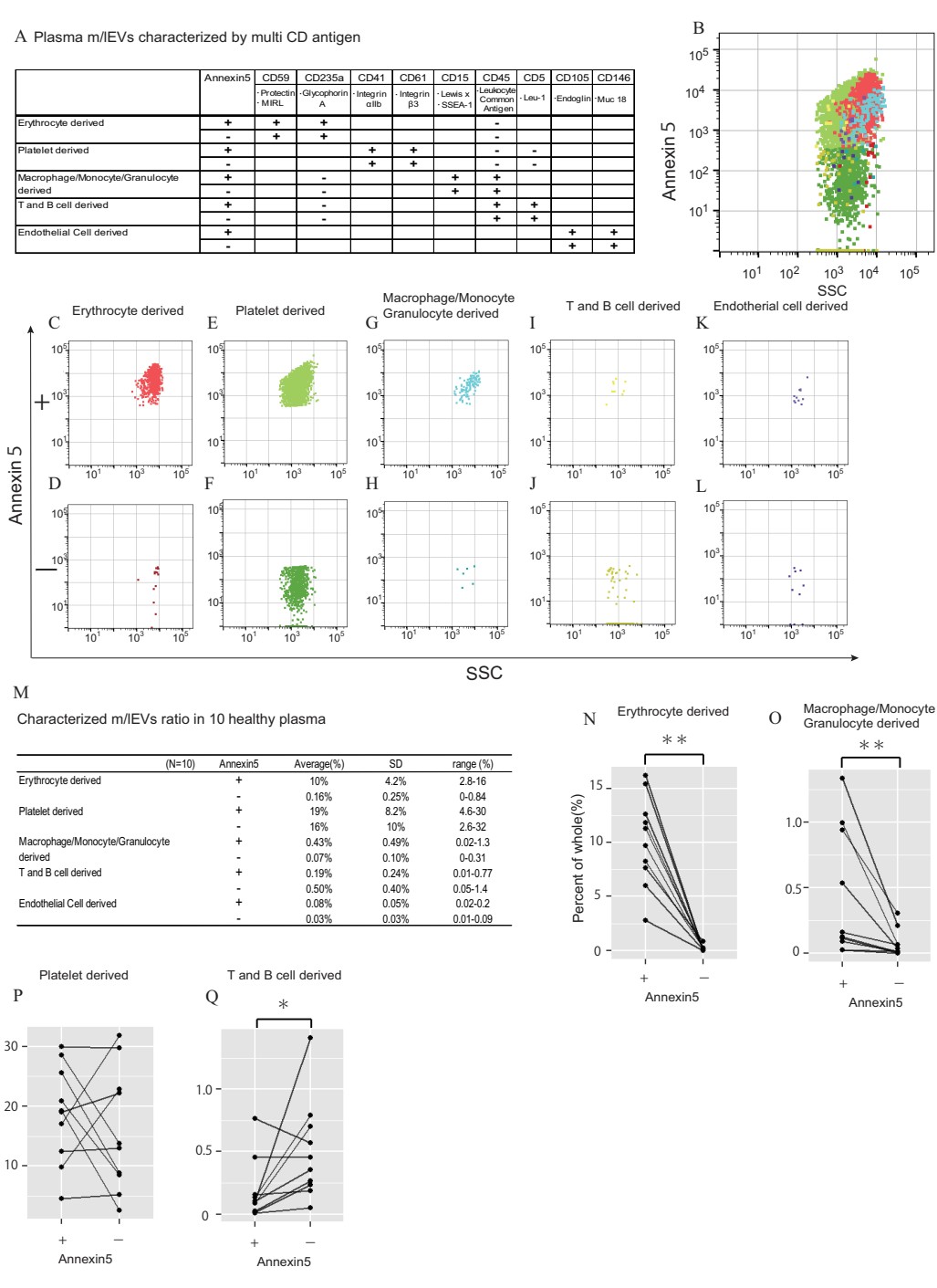

**Figure 4  Characterization of plasma m/lEVs by flow cytometry.** (A) Two specific surface antigens were used to characterize the m/lEVs from each source by flow cytometry. m/lEVs were characterized by comparison with five types of blood cells using surface antigens and Annexin V staining. (B–L) Representative dot plots (SSC vs. Annexin V). Each plot was classified by staining for surface antigens and Annexin V (C–L) and an overall plot summarizing the data (B) is shown. (M) Quantification of each m/lEV by flow cytometry analysis ($n = 10$ for human healthy plasma, % of total events). (N–Q) Comparison of Annexin V staining for erythrocyte-(N), macrophage-(O), platelet-(P) and T and B cell-(Q) derived m/lEVs from ten healthy plasma samples. Comparisons were performed using the Wilcoxon signed-rank test (*$p < 0.05$, **$p < 0.01$; a value of $p < 0.05$ was considered to indicate statistical significance).

staining for the m/lEVs corresponding to these five classifications (Figs. 4B–4L). We integrated these characterizations and assessed the distribution of EV classifications among ten healthy subjects (Fig. 4M). The results suggested that no major differences in the ratios of fractions in these ten subjects and thus these definitions may be used for pathological analysis.

We found that 10% and 35% of m/lEVs were derived from erythrocytes and platelets, respectively. However, only 0.5%, 0.6% and 0.1% of m/lEVs were derived from macrophages, leukocytes and endothelial cells, respectively suggesting that the ratio of m/lEVs of different cellular origins is dependent on the number of cells present in plasma (Fig. 4M). We also observed that most m/lEVs derived from erythrocytes and macrophages were Annexin V positive (Figs. 4N and 4O). By contrast, many Annexin V negative m/lEVs were identified among platelet- and T and B cell-derived m/lEVs (Figs. 4P and 4Q). Especially about erythrocyte-derived m/lEVs other studies have shown high percentages of phosphatidylserine-positive(:Annexin V positive) m/lEVs after red blood cell storage under blood bank conditions that these results are consistent (*Gao et al., 2013*; *Xiong et al., 2011*).

In general, it is known that microparicle in blood are known to be exposed to PS on the surface, which is verified by being positive by Annexin5 staining. We found that the degree of exposure of phosphatidylserine to the membrane surface was vary depending on the cell derived from annexin V staining. Thus, the characteristics of m/lEVs can be determined in detail by using AnnexinV and antigenicity. These results suggested that the degree of exposing PS are cell-type specific and that release mechanisms may differ among cell types.

## Characterization of urinary EVs by flow cytometry and enzyme activity assay

In urine, we first removed aggregated m/lEVs and residual THP polymers using labelled normal mouse IgG (Figs. S2C). By removing the THP polymer by DTT treatment, many immunological non-specific reactions in flow cytometry observation were eliminated, and the remaining non-specific reactions were completely excluded from the observed image by mouse IgG-positive gating-out (Figs. S2D). To characterize urinary m/lEVs, we used surface antigens detected by shotgun proteomic analysis including CD10 (neprilysin), CD13 (alanine aminopeptidase), CD26 (DPP4) and CD227 (MUC1) (Figs. 5A–5F). Many m/lEVs in the observation area were triple-positive for CD10, CD13 and CD26, but negative for Annexin V (Figs. 5B–5D, Figs. S6). Furthermore, MUC1-positive EVs were both Annexin V positive and negative in roughly equivalent frequencies (Figs. 5B, 5E and 5F). These results suggested that m/lEVs containing peptidases were released by outward budding directly from the cilial membrane of renal proximal tubule epithelial cells. The results of integrating these characterizations and the distribution of EV classifications among ten healthy subjects are shown (Figs. 5G–5I). These data indicated no major differences in the ratio among these populations, suggesting that our methodology was reliable for m/lEV analysis.

We next verified the CD26 peptidase enzyme activities of m/lEVs in plasma and urine from six individuals. We prepared three fractions: (i) "whole," in which debris were

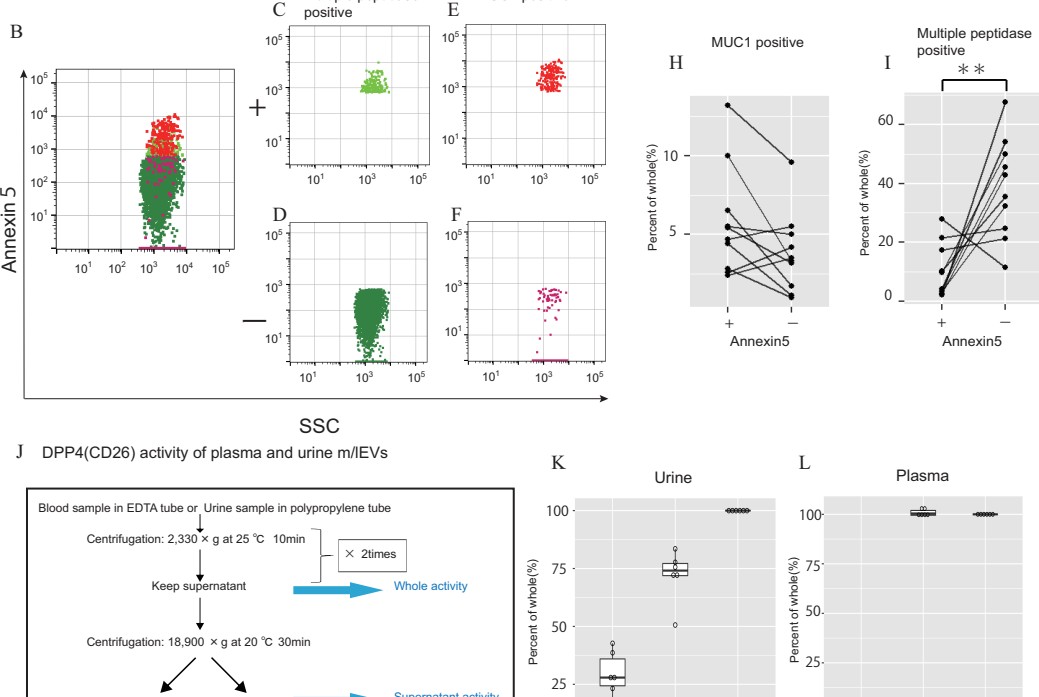

**Figure 5   Characterization of urinary m/lEVs by flow cytometry and CD26 enzymatic activity.** (A) Two kinds of m/lEVs in urine were characterized by using surface antigen and Annexin V staining. (B–F) Representative dot plots in the observation area (SSC vs Annexin V). Each plot was classified according to staining for multiple peptidases [CD10 (neprilysin), CD13 (alanine aminopeptidase) and CD26 (DPP4)] (C and D), and CD277 (MUC-1) (E and F) Events were further classified as Annexin V positive or negative. (G) Quantification of m/lEVs by flow cytometry analyses ($n = 10$ for male human healthy urine, % of total events). (H and I) Comparison of Annexin V staining for MUC1-positive (H) and multiple peptidase-positive (I) m/lEVs in the urine of ten healthy male human samples. Comparisons were performed using the Wilcoxon signed-rank test (*$P < 0.05$, ∗∗∗$P < 0.01$; a value of $p < 0.05$ was considered to indicate statistical significance). (J) DPP4 enzymatic activity was assessed for urinary m/lEVs. The workflow for fractionating m/lEVs by centrifugation is shown. (K and L) Quantification of DPP4 activity in urine (K) and plasma (L) fractions ($n = 6$ for human healthy male urine and plasma, % of whole). The percentage of enzyme activity measured for each fraction compared with whole activity is shown.

removed after low speed centrifugation, (ii) "m/lEVs" and (iii) "free (supernatant)" both of which were obtained via high speed centrifugation ($18,900 \times g$ for 30 min) (Fig. 5J). We found that more than 20% of DPP4 activity in whole urine was contributed by the EV fraction (Fig. 5K and Figs. S7). By contrast, there was no peptidase activity associated with

plasma m/lEVs (Fig. 5L). These results suggested that functional CD26 peptidase activity is present in m/lEVs in urine, which may be useful for pathological analysis.

## DISCUSSION

In this study, we analyzed m/lEVs using various analytic techniques and found the following four major results. First, it was possible to characterize m/lEVs using multiple surface markers. Second, m/lEVs bear functional enzymes with demonstrable enzyme activity on the vesicle surface. Trird, there are probability of differences in asymmetry of membrane lipids by derived cells. Finally, there was little variation m/lEVs in the plasma and urine of healthy individuals, indicating that our method is useful for identifying cell-derived m/lEVs in these body fluids.

We isolated m/lEVs from plasma and urine that were primarily 200–800 nm in diameter as shown by transmission electron microscopy. A large proportion of proteins detected in m/lEVs using shotgun proteomic analysis were categorized as plasma membrane proteins. Isolation of m/lEVs by centrifugation is a classical technique, but in the present study we further separated and classified the m/lEVs according to their cell types of origin by flow cytometry. The results indicated the validity of the differential centrifugation method (*Biro et al., 2003*; *Piccin et al., 2015*).

*Pang et al. (2018)* reported that integrin outside-in signaling is an important mechanism for microvesicle formation, in which the procoagulant phospholipid phosphatidylserine (PS) is efficiently externalized to release PS-exposed microvesicles (MVs). These platelet-derived Annexin V positive MVs were induced by application of a pulling force via an integrin ligand such as shear stress. This exposure of PS allows binding of important coagulation factors, enhancing the catalytic efficiencies of coagulation enzymes. We observed that 50% of m/lEVs derived from leukocytes and platelets were Annexin V positive, suggesting that release PS-positive m/lEVs during activation, inflammation, and injury. It would be interesting to further investigate whether the ratio of Annexin V positive m/lEVs from platelets or leukocytes was an important diagnostic factor for inflammatory disease or tissue injury.

In urinary m/lEVs, we identified aminopeptidases such as CD10, CD13 and CD26 which are localized in proximal renal tubular epithelial cells. The functions of these proteins relating to exocytosis were categorized by gene enrichment analysis. The cilium in the kidney is the site at which a variety of membrane receptors, enzymes and signal transduction molecules critical to many cellular processes function. In recent years, ciliary ectosomes—bioactive vesicles released from the surface of the cilium—have attracted attention (*Nager et al., 2017*; *Phua et al., 2017*; *Wood & Rosenbaum, 2015*). We also identified ciliary ectosome formation ESCRT complexes proteins (CHAMP; Tables S3 and S4) in proteomic analyses, suggesting that the possibility that these proteins were biomarkers of kidney disease. Because triple peptidase positive m/lEVs were negative for Annexin V, the mechanism of budding from cells may not be dependent on scramblase (*Wood & Rosenbaum, 2015*).

Platelet-derived m/lEVs are the most abundant population of extracellular vesicles in blood, and their presence (*Piccin, Murphy & Smith, 2007*) and connection with tumor

formation were reported in a recent study (*Zmigrodzka et al., 2016*). In our study, platelet-derived EVs were observed in healthy subjects and had the highest abundance of Annexin V-positive EVs. In plasma, leukocyte-derived EVs were defined as CD11b/CD66b- or CD15-positive (*Sarlon-Bartoli et al., 2013*). We characterized macrophage/monocyte/granulocyte- and T/B cell-derived EVs based on two specific CD antigens, and we confirmed that EVs derived from these cells were very rare. Importantly, there was little variation in the cellular origins of m/lEVs in samples from ten healthy individuals, indicating that this method was useful for identifying cell-derived m/lEVs. We plan to examine m/lEVs differences in patients with these diseases in the near future. Erythrocyte-derived EVs were also characterized by their expression of CD235a and glycophorin A by flow cytometry (*Ferru et al., 2014*; *Zecher, Cumpelik & Schifferli, 2014*).

We also characterized m/lEVs in urine. In kidneys and particularly in the renal tubule, CD10, CD13, CD26 can be detected in high abundance by immunohistochemical staining (website: The Human Protein Atlas). CD10/CD13-double positive labeling can be used for isolation and characterization of primary proximal tubular epithelial cells from human kidney (*Van der Hauwaert et al., 2013*). DPP4 (CD26) is a potential biomarker in urine for diabetic kidney disease and the presence of urinary m/lEV-bound DPP4 has been demonstrated (*Sun et al., 2012*). The presence of peptidases on the m/lEV surface, and their major contribution to peptidase activity in whole urine (*Sun et al., 2012*), may suggest a functional contribution to reabsorption in the proximal tubules. These observations suggested that the ratio of DPP activity between m/lEVs and total urine can be an important factor in the diagnosis of kidney disease.

MUC1 can also be detected in kidney and urinary bladder by immunohistochemical staining (website: The Human Protein Atlas). Significant increases of MUC1 expression in cancerous tissue and in the intermediate zone compared with normal renal tissue distant from the tumor was observed (*Borzym-Kluczyk, Radziejewska & Cechowska-Pasko, 2015*). In any case, MUC1-positive EVs are thought to be more likely to be derived from the tubular epithelium or the urothelium.

## CONCLUSIONS

Use of EVs as diagnostic reagents with superior disease and organ specificity for liquid biopsy samples is a possibility. This protocol will allow further study and in depth characterization of EV profiles in large patient groups for clinical applications. We are going to attempt to identify novel biomarkers by comparing healthy subjects and patients with various diseases.

## ACKNOWLEDGEMENTS

We thank lab members for reagents, discussions, and critical reading of the manuscript. We thank Edanz Group for editing a draft of this manuscript.

### Funding

This work was supported by research funding from LSI Medience Corporation, a Grant-in-Aid for Scientific Research from the Japan Society for the Promotion of Science (JSPS; grant numbers #25253041, #17H01550 and 15H04764). The funders had no role in study design, data collection and analysis, decision to publish, or preparation of the manuscript.

### Grant Disclosures

The following grant information was disclosed by the authors:
Japan Society for the Promotion of Science (JSPS): 25253041, 17H01550, 15H04764.
LSI Medience Corporation.

### Competing Interests

Ko Igami is a full-time employee of LSI Medience Corporation. Kazuyuki Kamioka is a full-time employee of LSI Medience Corporation. Ko Igami is also seconded to Kyushu Pro Search Limited Liability Partnership from LSI Medience Corporation. Kazuyuki Kamioka also serves as Kyushu Pro Search Limited Liability Partnership and LSI Medience Corporation and is the representative officer of Kyushu Pro Search Limited Liability Partnership. Ko Igami, Takeshi Uchiumi and Dongchon Kang have a patent application on this content. The institution to which the author belongs (Kyushu university and LSI Medience Corporation) hold the rights to the patent.

### Author Contributions

- Ko Igami conceived and designed the experiments, performed the experiments, analyzed the data, performed the computation work, prepared figures and/or tables, authored or reviewed drafts of the paper, application for research ethics committee, and approved the final draft.
- Takeshi Uchiumi and Dongchon Kang conceived and designed the experiments, performed the computation work, prepared figures and/or tables, authored or reviewed drafts of the paper, application for research ethics committee, and approved the final draft.
- Saori Ueda and Daiki Setoyama performed the experiments, analyzed the data, prepared figures and/or tables, and approved the final draft.
- Kazuyuki Kamioka analyzed the data, authored or reviewed drafts of the paper, and approved the final draft.
- Kazuhito Gotoh, Masaru Akimoto and Shinya Matsumoto performed the experiments, analyzed the data, authored or reviewed drafts of the paper, and approved the final draft.

### Ethics

The following information was supplied relating to ethical approvals (i.e., approving body and any reference numbers):

All studies were approved by the Institutional Review Board of the Kyushu University Hospital, Kyushu University (29-340).

## Patent Disclosures

The following patent dependencies were disclosed by the authors:

The authors have a patent application on this content:

- Patent 1: JP2018-109402/June 7, 2018, ''Hito ketsueki kara no maikurobeshikuru no bunri hōhō oyobi bunseki hōhō''.

- Patent 2: JP2018-109403/June 7, 2018, ''Hito nyō kara no maikurobeshikuru no bunri hōhō oyobi bunseki hōhō''.

## Data Availability

The supplemental data is available in the Supplementary Files.

## Supplemental Information

Supplemental information for this article can be found online at http://dx.doi.org/10.7717/peerj-achem.4#supplemental-information.

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
