# Peer review of "Characterization and function of medium and large extracellular vesicles from plasma and urine by surface antigens and Annexin V"

_PeerJ Analytical Chemistry, doi:10.7717/peerj-achem.4_

## Round 0.1 · original submission · Major Revisions

· Academic Editor

Major Revisions

I invite you to respond to the Referees' comments and revise your manuscript accompanied by a list to explain your revisions.

Reviewer 1 ·

Basic reporting

I really enjoyed reading this paper!!
It is very actual but also quite innovative and of interest because looking at explaining medium and large Extracellular Vesicles in healthy people
This paper is very well written and I believe it deserves to be published.
I have found only some minor spelling/errors and I have reported those here below.

ABSTRACT
This should be rewritten!

Need a sentence to introduce first EV then move on introducing m/lEV
However give also the size 200-800 nm here, not in the method.
please in the method need to mention that samples were drawn by healthy people
Were these blood donors? How do you ensure that they were healty? Did you measure other key parameters such as for example FBC and urine creatinine? if yes please provide these data in a Table

INTRODUCTION
line 51
add those more references:
Intern Med J. 2017 Oct;47(10):1173-1183. doi: 10.1111/imj.13550
Transl Res. 2017 Jun;184:21-34. doi: 10.1016/j.trsl.2017.02.001.
Blood Transfus. 2015 Apr;13(2):172-3. doi: 10.2450/2014.0276-14.
Acta Haematol. 2014;132(2):199.

line 65,
need to also comment on EV in VOD and on ET use these references:
Intern Med J. 2017 Oct;47(10):1173-1183. doi: 10.1111/imj.13550
Transl Res. 2017 Jun;184:21-34. doi: 10.1016/j.trsl.2017.02.001.

MATERIAL AND METHODS
when reporting industrial data or product the copyright sign " © "should be included
e.g Thermo Fisher, Wako etc

As above: healthy people how did you ensure this?
Were these blood donors? How do you ensure that they were healty?
Did you measure other key parameters such as for example FBC and urine creatinine?
if yes please provide these data in an additional Table

line 322 we next examined= rephrase it

DISCUSSION
line 353 delete "these cells are exposed to shear stress during blood flow and"

CONCLUSION
Delete first sentence: " We made...derived."

Experimental design

very well

Validity of the findings

very well

Additional comments

I really enjoyed reading this paper!!
It is very actual but also quite innovative and of interest because looking at explaining medium and large Extracellular Vesicles in healthy people
This paper is very well written and I believe it deserves to be published.
I have found only some minor spelling/errors and I have reported those here below.

ABSTRACT
This should be rewritten!

Need a sentence to introduce first EV then move on introducing m/lEV
However give also the size 200-800 nm here, not in the method.
please in the method need to mention that samples were drawn by healthy people
Were these blood donors? How do you ensure that they were healty? Did you measure other key parameters such as for example FBC and urine creatinine? if yes please provide these data in a Table

INTRODUCTION
line 51
add those more references:
Intern Med J. 2017 Oct;47(10):1173-1183. doi: 10.1111/imj.13550
Transl Res. 2017 Jun;184:21-34. doi: 10.1016/j.trsl.2017.02.001.
Blood Transfus. 2015 Apr;13(2):172-3. doi: 10.2450/2014.0276-14.
Acta Haematol. 2014;132(2):199.

line 65,
need to also comment on EV in VOD and on ET use these references:
Intern Med J. 2017 Oct;47(10):1173-1183. doi: 10.1111/imj.13550
Transl Res. 2017 Jun;184:21-34. doi: 10.1016/j.trsl.2017.02.001.

MATERIAL AND METHODS
when reporting industrial data or product the copyright sign " © "should be included
e.g Thermo Fisher, Wako etc

As above: healthy people how did you ensure this?
Were these blood donors? How do you ensure that they were healty?
Did you measure other key parameters such as for example FBC and urine creatinine?
if yes please provide these data in an additional Table

line 322 we next examined= rephrase it

DISCUSSION
line 353 delete "these cells are exposed to shear stress during blood flow and"

CONCLUSION
Delete first sentence: " We made...derived."

Reviewer 2 ·

Basic reporting

This manuscript describes the proteome analysis of medium and large extracellular vesicles (m/lEVs) from plasma and urine. I can not find out the importance and novelty of this study from the viewpoint of analytical chemistry, because authors used general analytical procedures and their findings does not directly lead to a new analytical method. Hence, I can not understand why this manuscript was submitted to an analytical chemistry journal. Please add the novelty, importance, and progress of this study in terms of analytical chemistry more concretely.

Experimental design

Although authors said that they collected 200-800 nm m/lEVs by differential centrifugation, EVs larger than 800 nm exist in your samples by the flow cytometric analysis (Figure 2). Authors used nanoparticle tracking analysis (NTA) (line 242) which is a useful instrument for size, size distribution and particle number analyses of the particles, please add the analytical results of your samples by NTA in the revised manuscript.

Validity of the findings

This study is important in the field of EVs.

Reviewer 3 ·

Basic reporting

no comment

Experimental design

no comment

Validity of the findings

no comment

Additional comments

The manuscript entitled “Characterization and function of medium and large extracellular vesicles from plasma and urine by surface antigens and Annexin V” shows that m/lEVs derived from plasma and urine can be characterized by flow cytometry analysis using some membrane proteins and their methodology can distinguish m/lEVs from each type of cells. Moreover, they revealed that DPP4 activity is present in m/lEVs from urine, but not m/lEVs from plasma. These methods are effective in quality control of m/lEVs from plasma and urine, however, the impact is lost by not indicating the clinical applications and I found no major flaws in this manuscript.

The following points should be addressed:
1. What is the aim of this study? If the authors claim that the novelty of this study is development of m/lEVs detection methods, some researcher already reported the EV detection methods using flow cytometer. Moreover, if the authors want to use this method for clinical applications, the authors need to clearly articulate why this study was undertaken (what clinical situations?).
2. The authors described “data not shown” in this manuscript, however, the authors should show the all data in manuscript.
3. Although the authors claimed that CD59 and CD235a double-positive and CD45-negative m/lEVs were classified as erythrocyte-derived m/lEVs etc…, there is no evidence shown in the manuscript. The authors should indicate CD59 and CD235a double-positive and CD45-negative m/lEVs are derived from erythrocyte using pure culture system (I mean the authors culture erythrocyte and purification of m/lEVs from this culture supernatant).
4. Why did the authors use Annexin V as a marker of m/lEVs? Please explain more details.

---

## Round 0.2 · Minor Revisions

· Academic Editor

Minor Revisions

Please respond to Reviewer 2's comment regarding experimental design.

Reviewer 1 ·

Basic reporting

Accept for publication

Experimental design

is well reported and clear

Validity of the findings

Are well reported and of interest

Reviewer 2 ·

Basic reporting

Thank you for your revisions based on my comments.

Experimental design

Authors mentioned that they characterized medium/large EVs (m/lEV) and their size ranges were from 200 to -800 nm in the manuscript. However, the size of more than half plasm EVs measured by NTA was less than 100 nm after centrifugation (Supplementary Fig.S1). How did you eliminate the effects of proteins in small EVs in shotgun proteomic analysis?

Validity of the findings

Although you added the importance of this study from the viewpoints of analytical chemistry, I couldn't understand the importance of this study even in the revised manuscript. For example, you found CD26 enzyme activity from urine samples in this study. Why is this important from an analytical chemistry viewpoint? Please explain the novelty and importance of this study in a simpler and easier way from an analytical chemistry viewpoint.

Reviewer 3 ·

Basic reporting

This revised version is much improved. The authors addressed almost all of my concerns.

Experimental design

This revised version is much improved. The authors addressed almost all of my concerns.

Validity of the findings

This revised version is much improved. The authors addressed almost all of my concerns.

Additional comments

This revised version is much improved. The authors addressed almost all of my concerns.

---

## Round 0.3 · accepted · Accept

· Academic Editor

Accept

Reviewer 2 was concerned about the presence of vesicles smaller than 100 nm in size, but I judged that their abundance was small and would not change the conclusions of this paper.

Reviewer 2 ·

Basic reporting

Thank you very much for your revisions. The revisions does not fit my requests. For example, I think you should analyze new sample because about half of the EVs have deteriorated before the NTA analysis. However, if you can not make additional revisions those were requested by my previous comments, no further improvements of the manuscript are expected. This revised manuscript would be accepted for publication.

Experimental design

no further comments

Validity of the findings

no further comments